# Water-Soluble Truncated Fatty Acid–Porphyrin Conjugates Provide Photo-Sensitizer Activity for Photodynamic Therapy in Malignant Mesothelioma

**DOI:** 10.3390/cancers14215446

**Published:** 2022-11-05

**Authors:** Sam Bonsall, Simeon Hubbard, Uthaman Jithin, Joseph Anslow, Dylan Todd, Callum Rowding, Tom Filarowski, Greg Duly, Ryan Wilson, Jack Porter, Simon Turega, Sarah Haywood-Small

**Affiliations:** Biomolecular Sciences Research Centre, Sheffield Hallam University, City Campus, Howard Street, Sheffield S1 1WB, UK

**Keywords:** photodynamic therapy, mesothelioma, apoptosis, necrosis

## Abstract

**Simple Summary:**

Malignant mesothelioma is a rare, fatal cancer with limited treatment options. Photodynamic therapy is an emerging treatment option which uses a drug (known as a photosensitizer) and a specific type of light to selectively destroy cancerous cells. We have synthesized new photosensitizing drugs by linking organic chemical structures (variable length fatty acid chains) to existing photosensitisers. These new photosensitizing drugs were characterized, and their cancer-killing capabilities were also assessed in malignant mesothelioma cells in the laboratory. We have found that one of the newly synthesized photosensitisers (C5SHU) appeared to cause a significant amount of cell death in the malignant mesothelioma cells in combination with light. C5SHU-induced cell death did not occur in the absence of light. Therefore, we propose that this photosensitizer—C5SHU—is a suitable anti-tumour drug candidate in this hard-to-treat cancer.

**Abstract:**

Clinical trials evaluating intrapleural photodynamic therapy (PDT) are ongoing for mesothelioma. Several issues still hinder the development of PDT, such as those related to the inherent properties of photosensitizers. Herein, we report the synthesis, photophysical, and photobiological properties of three porphyrin-based photosensitizers conjugated to truncated fatty acids (C5SHU to C7SHU). Our photosensitizers exhibited excellent water solubility and high PDT efficiency in mesothelioma. As expected, absorption spectroscopy confirmed an increased aggregation as a consequence of extending the fatty acid chain length. In vitro PDT activity was studied using human mesothelioma cell lines (biphasic MSTO-211H cells and epithelioid NCI-H28 cells) alongside a non-malignant mesothelial cell line (MET-5A). The PDT effect of these photosensitizers was initially assessed using the colorimetric WST-8 cell viability assay and the mode of cell death was determined via flow cytometry of Annexin V-FITC/PI-stained cells. Photosensitizers appeared to selectively localize within the non-nuclear compartments of cells before exhibiting high phototoxicity. Both apoptosis and necrosis were induced at 24 and 48 h. As our pentanoic acid-derivatized porphyrin (C5SHU) induced the largest anti-tumor effect in this study, we put this forward as an anti-tumor drug candidate in PDT and photo-imaging diagnosis in mesothelioma.

## 1. Introduction

Photodynamic therapy (PDT) consists of the use of a tumor-specific photosensitizer (PS) and laser irradiation to induce the production of reactive oxygen species in cancer cells [1,2,3]. Molecular PDT therapeutics can be used as antimicrobials, as cancer targeting therapeutics, and in the treatment of skin problems such as vitiligo or psoriasis [4]. PDT is an emerging therapeutic option for a range of cancer types; this alternative treatment modality may both prolong the life of patients with inoperable cancers and improve quality of life [5]. Compared to more traditional treatments (chemotherapy and radiotherapy), PDT is less invasive and presents fewer adverse effects [6]. This is due to its more tumor-specific mode of action and the controllable toxicity of components; the components of PDT should be individually harmless but synergistically confer tumor destruction.

The photoactivated PS generates reactive oxygen species (ROS) either through sensitizing biomolecules to radicals/ions (type I reaction) or via the generation of singlet oxygen (type II reaction) [1]. The discrete generation of intracellular cytotoxic agents induces a cascade of cell death mechanisms, which lead to apoptosis, necrosis, autophagy, damage of tumor vasculature, and the provocation of an immune response. These factors make PDT most effective against localized, solid tumor-forming malignancies [7].

Malignant pleural mesothelioma (MPM) is an incurable cancer of the lungs primarily associated with occupational asbestos exposure. MPM has a limited median survival time (<12 months) and long latency period, which can reach up to 70 years in some cases [8]. Treatment options for MPM generally follow a multi modal approach, with chemotherapy, radiotherapy and surgery constituting the three main treatment areas, which are administered on a case-by-case basis depending on the patient and the severity of the disease and symptoms [9,10,11] However, data generally suggest that these treatment options do not significantly improve overall survival for MPM patients and mainly provide a palliative benefit. PDT presents a treatment modality which could be administered intraoperatively and combined with pleurectomy or pneumonectomy to potentially improve MPM patient outcomes [12,13].

In the clinic several photosensitizers have been used in the treatment of mesothelioma, based around the tetrapyrrole scaffold: hematoporphyrin derivatives, Photofrin for example, Foscan, and HPPH [14,15,16]. The hematoporphyrin derivatives are first-generation PS constructed around the porphyrin scaffold, the second-generation Foscan and HPPH PS are designed around the chlorin scaffold where one double bond of the porphyrin is reduced providing a strong, red-shifted absorbance at around 660 nm, compared to a weaker absorbance at 630 nm in the porphyrin derivatives [17,18].

Over the last 30 years, the most common PS used to treat mesothelioma in the clinic is Photofrin. Photofrin is typically given at a dose of 2 mg/kg, 24 h preoperatively and the PS is excited intraoperatively using 630 nm laser light. With the development of modern surgical treatments for MPM the addition of photodynamic therapy as multi modal treatment became an interesting option, both Photofrin and Foscan have been used to develop safe intraoperative PDT treatments [19,20]. These intraoperative PDT treatments are currently at a stage where they can be applied to new PS including the second-generation PS HPPH coupling of these newer PS with the advanced surgical techniques and laser tracking technology [21,22]. Hematoporphyrin dimers and oligomers, providing the porphyrin scaffold as a starting point for developing PS molecules. Second-generation molecules including the chlorins Foscan and HPPH have now been included in clinical studies [23]. Current PS development focuses on the synthesis of third generation PS, conjugates between targeting biological molecules and PS chromophores. Here, PSs based on the conjugation of truncated fatty acids and a porphyrin are introduced, changing the length of the truncated fatty acid will change the molecular recognition profile of the molecules by altering hydrophobic surface area/lipophilicity leading to a targeted delivery.

The porphyrin-based PS in this work were designed with truncated fatty acid chains conjugated to the phenyl rings in the 5, 10, 15 and 20 carbon positions around the porphyrin core. The fatty acid truncations are connected to the porphyrin’s phenyl rings via an ether linkage four, five, and six carbons from the carbonyl carbon of a straight chain fatty acid giving porphyrin C5SHU, C6SHU, and C7SHU, respectively. The fatty acid-type conjugation is designed to allow cell permeability through diffusion, low-density lipoprotein (LDL) transport or fatty acid specific transport protein activity [24,25]. Selectivity for diseased cell types over healthy cells will provide differences in membrane behavior, endocytosis, and exocytosis in cancerous cells. To understand the molecular recognition caused by these changes in structure of the PS we considered those changes in terms of molecular descriptors regularly used in medicinal chemistry. The PSs C5SHU, C6SHU, and C7SHU vary by four CH_2_ groups 5 to 6, 6 to 7 providing an additional; 75 Å^2^ of SASA, and 68 Å^3^ of volume per one carbon extension. These changes in carbon chain length give SlogP values of 13.9, 15.4, and 17 for molecules C5SHU, C6SHU, and C7SHU, respectively, highlighting the increase in lipophilicity as the truncated fatty acids get longer. Considering previous experimental measurements relating SASA to hydrophobic interaction the increase of SASA of 75 Å^2^ per one carbon extension of the hydrocarbon chain increases the potential for hydrophobic interactions by 20 kJ mol^−1^ [26,27,28,29]. This change in molecular recognition can also be described by the change in hydrophobic surface site interaction points (SSIPs) that can be used to describe the solvation properties of the fatty acid chain sections of the PS. Calculating the change in SSIPs that corresponds to a one carbon extension of the hydrocarbon chains gives a change in eight hydrophobic SSIPS per molecule, demonstrating the change in molecular recognition profile as the carbon chain is extended [30,31,32].

The aim of this initial report was to assess the PDT effect of a panel of three porphyrin derivatives in the human MSTO-211H and NCI-H28 mesothelioma cell lines. These cell lines were chosen as models due to their varying characteristics. MSTO-211H is of biphasic origin, consisting of epithelioid and sarcomatoid cell types. NCI-H28 is from a stage 4 mesothelioma consisting of purely epithelioid cells [33,34,35]. The human mesothelial cell line, Met-5a was used as a model of non-malignant mesothelial cells to evaluate the tumor-selectivity of this PDT effect [12,13].

## 2. Materials and Methods

**Commercial reagents**. Cell death staining reagents were all acquired in the FITC Annexin V Apoptosis Detection Kit with PI (BioLegend, London, UK). WST-8 (2-(2-methoxy-4-nitrophenyl)-3-(4-nitrophenyl)-5-(2, 4-disulfophenyl)-2H-tetrazolium) reagent was part of a pre-made solution in the Cell Counting Kit-8 (Sigma-Aldrich, Dorset, UK).

### 2.1. Optical Spectroscopy

UV/vis absorbance coefficient measurement in DMSO. Volumetric solutions of porphyrins 2–7 (0.5–2 mM) in DMSO were diluted to give working solutions (2.5 × 10^−2^–1 × 10^−1^ mM). For each porphyrin aliquots of working solutions were added to a quartz cuvette containing DMSO (2 mL), the absorption spectra was recorded from 400-800 nm after each addition. Absorbance values at the λ_max_ of the Soret band and four Q-bands were plotted as a function of PS concentration for use in the calculation of molar absorption coefficient.

UV/vis absorbance spectroscopy in DMSO, water and 1 M HCl. For each of the porphyrins 5, 6, and 7, 5 mg, was dissolved in 30 µL of 1 M NaOH H this was diluted to 5 mL with distilled to give stock solutions of concentrations 0.926 mM, 0.88 mM and 0.77 mM, respectively. In a quartz 96 well plate, 20 µL was added to a well and diluted to a volume of 320 µL with either DMSO, water or 1 M HCl.

### 2.2. Synthesis 

See Appendix A.

#### Porphyrin PS Stock Solution Preparation

Porphyrin PS stock solutions 4, 5, and 6 (Figure 1) were prepared by dissolving a 5 mg aliquot in 500 μL of 1 M NaOH. This was added to 9.5 mL of complete media to prepare a 500 μg/mL stock solution. Stock solutions were filter sterilized using a Millipore PES membrane filter (Merck, Watford, UK). Working solutions were prepared immediately before PDT experiments and consisted of six treatments: an untreated control, vehicle control, and four porphyrin concentrations (5, 10, 15, and 20 μg/mL).

### 2.3. Cell Culture

The human mesothelioma cell lines, MSTO-211H and NCI-H28, and the non-cancerous mesothelial cell line, MET-5a (ATCC, Manassas, VA, USA) were grown and routinely maintained in RPMI 1640 media (Gibco, London, UK). The complete growing media was supplemented with 2 mM L-glutamine (Gibco), 10% (*v*/*v*) fetal bovine serum (Gibco) and 1% (*v*/*v*) Penicillin and Streptomycin (Lonza, Bath, UK). The cell culture flasks were incubated at 37 °C and maintained in a 5% CO_2_:95% air humidified atmosphere. All cell lines passed routine mycoplasma testing using the MycoAlert^TM^ Mycoplasma Detection Kit (Lonza Group Ltd., Basel, Switzerland).

#### 2.3.1. Cell Viability Assays Using WST-8

Cells were detached using a trypsin/EDTA solution (Lonza). Trypan blue (0.4% *v*/*v*) exclusion was used to count viable cells before seeded into 96-well plates (Nunc) at a density of 5 × 10^3^ cells per well in 200 μL of complete RPMI. Cell culture plates were seeded approximately 24 h before PDT. Media was aspirated and replaced with equal volumes of the respective PS treatment. For the purpose of background subtraction, PS treatments were also added to wells without cells. Following 3 h of incubation (37 °C, 5% CO_2_:95% air), plates were irradiated for 15 min on a 15 W light box (UVP, Cambridge, UK). Experiments performed parallel to dark-treated control plates.

Cell viability was assessed by the colorimetric WST-8 assay. Following 24 h of incubation (37 °C, 5% CO_2_:95% air), 10 μL of WST-8 was added to each cell-containing well of the 96-well plates. The plates were incubated for 3 h (37 °C, 5% CO_2_:95% air) and then analyzed at 450 nm using a Clariostar Plate Reader (BMG Labtech, Ortenberg, Germany). Corrected absorbance was calculated in each case by subtraction of the media only controls. Finally, percentage cell viability was expressed relative to untreated control values.

#### 2.3.2. Cell Death Analysis Using Annexin V-FITC/PI

Cells were harvested in accordance with methods previously described and seeded into 24-well plates (Nunc, London, UK) at a density of 2 × 10^5^ cells per well in 1000 μL of complete RPMI; the culture plates were seeded approximately 24 h before PDT. Cell treatment and irradiation was performed in accordance with methods previously described.

Annexin V-FITC/PI staining and cell death analysis was performed at both 24- and 48 h time points following PDT (representative flow cytometry dot-plots from Annexin V-FITC/PI-stained cells in Appendix A). Non-adherent cells were harvested through the removal of culture medium and adherent monolayers were trypsinized and returned to the removed media. Harvested cells were removed from media via centrifugation (5 min, 400× *g*) (VWR Mega Star, Leicestershire, UK) and resuspended in Annexin V Binding Buffer. Following a second centrifugation, Annexin V Binding Buffer was discarded, and cell pellets were resuspended in the residual volume. Cells were stained with Annexin V-FITC (5 μL) and PI (10 μL) and incubated at room temperature for 15 min protected from light. 200 µL of Annexin V Binding Buffer was added prior to immediate analysis using a BD FACSCalibur (BD Biosciences, Wokingham, UK) and data from 10,000 events was collected. Flow cytometry results were analyzed using FlowJo™ v10.8 Software (BD Life Sciences). Quadrants consisted of Viable (Annexin V-FITC^−^/PI^−^), early stage apoptotic (Annexin V-FITC^+^/PI^−^) and late-stage apoptotic/necrotic (Annexin V-FITC^+^/PI^+^) populations.

#### 2.3.3. Data Analysis

Data points were expressed as the mean ± standard deviation (SD) of three independent experiments. Statistical analysis was performed using a two-way Analysis of Variance (ANOVA) with Bonferroni post hoc tests in GraphPad Prism (San Diego, CA, USA). Data points where *p* ≤ 0.05 were statistically significant in these experiments.

#### 2.3.4. Cellular Localization

Cells were harvested in accordance with methods previously described and seeded into 24-well plates (Nunc, London, UK) at a density of 25,000 treated with porphyrin working solutions made up from the porphyrin PS stock solutions and incubated for 24 h. The cells were visualized using a Cytation 5 (Agilent Technologies, Cheadle, UK) high content microscopy platform at 20× magnification. 

### 2.4. Calculation of Fatty Acid Solvation Properties

The SSIPTools suite of software, made available by the Hunter group, was used to describe the solvation properties of the fatty acid chain sections [30,31]. A range of chain lengths, from 1 to 12, with terminal OMe moieties were calculated. Structurally optimized mol2 files were converted into cml format and processed in NWChem [32]. SSIPs were assigned to the optimized structures using MEPS data by foot printing the cube data onto the further optimized cml structures. The complete set of SSIPs for each chain length was analyzed selecting SSIPs that correspond to the solvophobic region of the FGIP of water, this provided an average value of change in SSIP able to take part in solvophobic interactions per one carbon chain length extension in the PS which in this case correlates to hydrophobic interactions.

## 3. Results

Three PS, porphyrins conjugated to truncated fatty acid chains were synthesized from 5,10,15,20-Tetrakis(3-hydroxyphenyl)-21H,23H-porphyrin, 1 [36]. Bromo-esters with acid chain lengths of four, five and six carbons from the carbonyl carbon of the ester were connected to the hydroxy functionality of 1 in a Williamson ether synthesis to give porphyrins 2, 3 and 4, respectively, (Figure 1) in yields of 46, 36, and 54%. The tetrakis-ester porphyrins 2, 3, and 4 were hydrolyzed with KOH in methanol to give the target porphyrin tetrakis acids C5SHU, C6SHU, and C7SHU in yield of 68, 98, and 94%, respectively. The ionizable nature of the tetrakis carboxylic acids allows the porphyrins to be solubilized in aqueous solutions the porphyrins can be dissolved in a minimum volume of 1 M NaOH then diluted with water or buffered media to give a stock solution for spectroscopy or cell-culture. There is concern that porphyrins that are equilibrated in a stacked state will function differently in terms of cellular uptake and PS behavior to those that are well-solvated. The PS were observed in three different solvation systems to explore the stacking behavior. DMSO was firstly used as a negative control to ensure the PS is fully solvated and unable to stack. Distilled water was used to visualize stacking, and finally, the third condition used 1 M HCl as a positive control to forcibly induce stacking in the PS. Once dissolved in an aqueous solution, the absorption spectra of the porphyrins C5SHU, C6SHU, and C7SHU in water was compared to that of the solvated porphyrin recorded in polar organic solvent DMSO and the porphyrin forced to stack in 1 M HCl (Figure 1). The spectra of C5SHU in water has a sharp Soret band like that of the solvated C5SHU in recorded DMSO, the spectra of C6SHU and C7SHU in water have Soret bands that are broader than those recorded in DMSO spectra but not as the broad Soret bands seen in the stacked porphyrins [37,38,39].

The fluorescent behavior of the porphyrin PS arises from the 18-electron aromatic π system modulated by the additional carbon-carbon double bonds in the carbon 2 to 3 and carbon 12 to 13 positions around the ring. This fluorescence provides the opportunity to observe the uptake of the PS in the MSTO-211H, NCI-H28, and Met-5a cell lines being used in this study using a fluorescence microscopy approach. Treatment of a cell line followed by a 24 h incubation and visualization using a high-content microscopy platform allowed detection of the fluorescent PS in the chosen cell line. The C7SHU PS can be seen to localize in the cytosolic region of the MSTO-211H cell line in Figure 2 highlighting an absence of nuclear uptake of the PS. Using the high-content approach, it was possible to confirm that similar uptake and localization was observed in all three cell lines with each porphyrin (Figure 3).

For these experiments, a high-content approach was taken for the initial assessment of photosensitizer activity, allowing the irradiation of multiple PS conditions and multiple cell lines simultaneously using a white light source. Cell viability was assessed using the WST-8 assay quantifying viable cells through the metabolism of a water-soluble tetrazolium salt to a formazan dye. This was performed to establish the initial PDT effect of each PS in differing cell lines. Individually the MSTO-211H, NCI-H28, and Met-5a cell lines were incubated with various PS concentrations (5, 10, 15, and 20 μg/mL) in light and dark treatment conditions at both 24- and 48 h intervals post treatment. This was performed in the presence of untreated and vehicle controls, with cell viability being presented as a percentage of the untreated cells.

The MSTO-211H cell line demonstrated a sensitivity to all three PS C5SHU, C6SHU and C7SHU in both the 24 h and 48 h treatments (Figure 4). The C5SHU demonstrated a strong PDT reduction in cell viability from the 5 μg/mL treatment at the 48-h time point (22 vs. 100% viability in light and dark treatment conditions, respectively). However, further confirmation is needed due to the low viability in vehicle control treatments (≤35%) and the substantial dark toxicity at 15 and 20 μg/mL concentrations (30 and 26%, respectively after 48 h). C6SHU gave a defined PDT effect from the 5 μg/mL treatment at the 48-h time point (38 and 119% viability in light and dark treatment conditions, respectively) but not at the 24 h time point.

The NCI-H28 cell line displayed some sensitivity to the C5SHU and C6SHU porphyrins, which also appeared to potentiate at the 48-h time point (Figure 5). C5SHU demonstrated the most significant difference between cell viability in light and dark treatment conditions from the 5 μg/mL treatment (16 and 116% viability, respectively, at the 48 h time point).

The non-cancerous cell line, Met-5a demonstrated a similar C5SHU sensitivity to NCI-H28. Cell viability in light treatments plateaued to approximately 28% 24 h after the 5 μg/mL treatment. This was potentiated at the 48 h time point (approximately 11% after the 5 μg/mL treatment). A similar PDT effect was observed with C7SHU treatment, but to a lower potency. A reduction to 66 and 30% viability was observed in 5 μg/mL light treatments at 24- and 48-h time points, respectively (Figure 6).

In summary, the WST-8 cell viability permitted the observation of light toxicity in all three cell lines (MSTO-211H, NCI-H28 and Met-5a) with all three PS (C5SHU, C6SHU and C7SHU), but further confirmation is needed. Therefore, the mode of cell death was assessed by flow cytometry using the Annexin V-FITC/PI assay. Both the MSTO-211H and NCI-H28 cell lines were assessed 24- and 48 h following PDT. Viable, early stage apoptotic and late-stage apoptotic/necrotic cell populations were determined by FITC-labelled Annexin V and PI staining. Cell viability can provide an initial measure of dose response. Table 1 displays the normalized porphyrin derivative concentrations that gave a 50% reduction in viable cell populations (EC_50_).

C5SHU demonstrated a potent PDT effect in both NCI-H28 and MSTO-211H cells (EC_50_ values of 7.8 and 7.7 μg/mL, respectively 24 h time point) (Table 1). In light treatment conditions, cell death increased in a dose dependent manner. At the 10 μg/mL treatment, NCI-H28 displayed predominantly necrotic cell death (16% apoptotic and 53% necrotic cell populations) and MSTO-211H predominantly apoptotic (49% apoptotic and 34% necrotic cell populations). Cell death was further potentiated at the 48 h time point (Figure 7). In dark treatment conditions, the NCI-H28 cell line displayed no significant dark toxicity, with viability remaining unchanged with porphyrin dose (approximately 95% viable cells). Although the MSTO-211H cell line displayed minimal overall dark toxicity, significant dark toxicity was observed with higher porphyrin doses at the 48 h time point; 15 and 20 μg/mL treatments displayed 63.6 and 46.1% viable cells, respectively (Figure 7). The Met-5a cell line displayed no significant cell death and negligible dark toxicity at the 24 h time point for all dose concentrations (Figure 7). Negligible dark toxicity was also observed at the 48 h time point, and significant decreases in the population of viable cells could be observed at all doses, with the predominant mode of cell death being necrosis rather than apoptosis (Figure 8).

The C6SHU PS did not induce a potent PDT effect in NCI-28 (IC_50_ value of >20 μg/mL) (Table 1). A substantial reduction in NCI-H28 viability was only observed at the 20 μg/mL light treatment (59% viable cells). A PDT effect was observed in MSTO-211H; the 20 μg/mL light treatment reduced the viable cell population to 20%, presenting predominantly necrotic cell death (23% apoptotic and 50% necrotic cell populations) giving an EC_50_ of 14 μg/mL (Figure 9 and Figure 10; Table 1). Activity in the Met-5a cells was like the NCL-H28 line but noticeably less active than in the MSTO-211H and at both time points. Like C6SHU, C7SHU (Figure 11) was less potent in the NCI-H28 cell line (EC_50_ values of > 20 μg/mL). At the 10 μg/mL treatment, NCI-H28 displayed a high proportion of viable cells (75% at the 48 h time point) (Figure 12). The MSTO-211H cell line was substantially more receptive to C7SHU (EC_50_ value of 9.6 μg/mL) (Table 1).

## 4. Discussion

The porphyrins C5SHU, C6SHU, and C7SHU were synthesized and then characterized using spectroscopic and mass-spec techniques to confirm their identity and purity (see NMR spectra, mass spectrometry data and UV/vis absorbance coefficient measurements in Appendix A). A successful vehicle system that allowed the porphyrins to be solubilized in aqueous solutions by dissolving in a small volume of 1 M NaOH(aq); this meant the PS could be reproducibly used in cancer biology experiments. An initial spectroscopic investigation of supramolecular stacking driven by electrostatic π orbital interactions in the porphyrins (Figure 1) suggested that the C5SHU PS showed the least evidence of stacking in aqueous solutions [38]. The fluorescence of PS in the three cell lines was observed using high-content microscopy, highlighting the cytosolic uptake of all three PS in all three cell lines and the absence of nuclear uptake. The lack of nuclear uptake observed is considered beneficial to maintain the major site of cellular regulation that ultimately preserves genomic integrity, especially in non-malignant cells [40]

The ability of all three photosensitizers (C5SHU, C6SHU, and C7SHU) to induce light-mediated cytotoxicity and mode of cell death induced was evaluated via a combination of methods. The WST-8 cell viability permitted the observation of light toxicity in all three cell lines (MSTO-211H, NCI-H28, and Met-5a) with all three PS (C5SHU, C6SHU, and C7SHU). The Annexin V-FITC/PI assay confirmed and provided further evidence of specific cell death populations in the light conditions. The five-carbon porphyrin derivative (C5SHU) was shown to be the most effective PS as demonstrated by the Annexin V-FITC/PI assay. The MSTO-211H displayed significant dark toxicity at the 48 h 15 and 20 μg/mL treatments (*p* ≤ 0.01 and *p* ≤ 0.001, respectively), this was the most dose-sensitive cell line/PS combination in light treatments. The NCI-H28 cell line gave the most archetypal response to C5SHU, demonstrating a light-specific induction of cell death; in dark conditions, the proportion of viable cells (approximately 95%), was not impacted by C5SHU concentration.

The Annexin V-FITC/PI assay displayed fewer instances of dark toxic effects compared to the WST-8 assay and demonstrated substantially greater consistency between experimental repeats. In the cell death analysis of C5SHU, NCI-H28 displayed highly necrotic cell death whereas MSTO-211H underwent predominantly apoptotic cell death. This can be attributed to NCI-H28 originating from a drug-resistant mesothelioma, the pre-treatment of this cell line may have given rise to more anti-apoptotic mutations [12,13,34]. The induction of necrotic cell death is necessary for the prolonged, secondary anti-tumor effects of PDT. Intracellular content released via necrosis may lead to the provocation of an immune response in vivo, which increases the curative potential of PDT. Indirect tumor killing via an immune response is advantageous for tumors with apoptosis-suppressing genetic mutations [23].

In summary, C5SHU was the most efficacious PSs identified in this work. The PDT effect of C5SHU observed in the non-cancerous Met-5a cell line suggests that the PSs also accumulate in healthy tissue. A small amount of PDT activity was observed in the non-cancerous Met-5a cell line, suggesting further physicochemical modifications may be necessary to improve the bioavailability of the PSs. However, PDT activity can be optimized via the conditions of light exposure, total light dose, and fluence rate [41]. It is also important to acknowledge that immortalized ‘normal’ cell lines have also been reported to have genomic heterogeneity levels comparable to those of transformed cell lines [42]. The biphasic MSTO-211H cell line displayed the greatest variability between experiments, suggesting that variation in the populations of epithelioid and sarcomatoid cell types within the MSTO-211H cell line may affect either PDT efficacy or PS uptake. This conjoined with more effective methods of photoactivation for example an LED system with a defined wavelength and fluence rate would provide a more standardized photo-activation. While this preclinical report demonstrates the direct tumor killing of these porphyrin derivatives, it does not elucidate their potential anti-vascular or immunogenic means of tumor targeting. By exciting PS-treated cells specifically at wavelengths corresponding to the Soret or Q bands of the PSs, their affinity for various cellular targets (nucleus, cell membrane, and mitochondria) may also be established.

The C5SHU PS demonstrated the highest efficacy in all the cell lines investigated, in the stacking studies this PS demonstrated the least evidence of stacking in aqueous solutions. If we combine this observation with our initial EC_50_ measurements and the possibility of both shortening and extending the truncated fatty acid substituents, then it is possible to visualize that a quantitative structure activity relationship (QSAR) could be produced for an expanded group of PS. The QSARs produced will give a high-quality understanding of which of the available PS is most efficacious and the direction in which the development of PS and use in treatment can take.

## 5. Conclusion

An initial comparison of the molecular description of the three PS molecules has been made in terms of molecular volume, fatty acid chain length, surface area, and hydrophobicity. The physical behavior of the PS has been initially probed using a measure of aggregation induced by stacking of the PS molecules. Considering the measure of PS activity in the cell lines used, with the initial EC_50_ values, it is possible to differentiate between the PS activity of all the PS C5SHU, C6SHU, and C7SHU in an individual cell line. The same EC_50_ values provide a difference in activity of an individual PS in the three cell lines each with different origins and phenotypic characteristics. This tells us that the C5SHU molecule with the shortest chain length and lowest hydrophobic surface area is the most efficacious PS, and it targets the MSTO-211H cells most effectively. This provides a range of parameters that we can correlate with activity in future work to better synthesize PS and target these PS to MPM tumor cells as part of a treatment.

## Data Availability

No new data were created or analyzed in this study. Data sharing is not applicable to this article.

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
