# Peer review of "Water-Soluble Truncated Fatty Acid–Porphyrin Conjugates Provide Photo-Sensitizer Activity for Photodynamic Therapy in Malignant Mesothelioma"

_cancers, 2022, doi:10.3390/cancers14215446_

Round 1

Reviewer 1 Report

The manuscript submitted by Bonsall et al, have the synthesis, photophysical and photobiological properties of three porphyrin-based photosensitisers conjugated to truncated fatty acids (C5SHU to C7SHU). Auhtors claimed that the photosensitisers exhibited excellent water solubility and high PDT. Auhtors have also studied these photosynthesisers in different human mesothelioma cell lines and finally put forward them as anti-tumor drug candidate in PDT and photo-imaging diagnosis in mesothelioma. Though this is an interesting topic to be explored at the current scenario, unfortunately, the manuscript submitted falls considerably short of its promise. My suggestions for improving the manuscripts are given below.

1.       Line 219, Authors should discuss the reason of having different spectra in different solvents

2.       Line 236, Author should use some nucleus staining dye to show the difference between nuclear stain and cytoplasmic stain and then can only exclude the absence of nuclear uptake

3.       Line 238, Authors did not discuss about the difference in staining in different cells and different PS

4.       The figure 3 quality is very poor

5.       Line 257, This statement is so confusing, as authors reported that C5SHU shows strong reduction in cell viability however, the figure does not depict it

6.       Figure 4,5, and 6 representation is very poor, and data presented in the figure and discussed in results section so confusing e.g., Figure 4, C5SHU and C6SHU, how come the cell viability at 5 ug/ml increased to zero or no treatment?

7.       Figure 7 and 8, authors did not present the FACS images of Annexin and PI staining which creates doubts to the graphs presented

8.       In supplementary authors should also present the mas spectra of compounds synthesized to show the purity of compounds

This is an important topic however, I regret not being able to provide more positive comments, but I believe at this form the manuscript is not mature enough to be published in cancer journal.  

Reviewer 2 Report

With great interest I have read the manuscript titled "Water soluble truncated fatty acid-porphyrin conjugates provide photo-sensitizer activity for photodynamic therapy in malignant mesothelioma." submitted by Bonsall and co-workers.

As an experienced general thoracic surgeon I have been working with both, endoscopic and intraoperative PDT for 20 years, treating esophageal cancer, lung cancer and malignant pleural mesothelioma, respectively. Aside from that I have published several papers dealing with the diverse impact of PDT in cancer patients so I still know about the various advantages and drawbacks of endoscopic and intraoperative PDT in the clinical setting.

With regard to my biochemical/ photochemical understanding of PDT I really appreciate the preliminary results obtained in the in-vitro setting in this present manuscript. In my opinion these performed modifications of the photosensitizer should be intensified and continued in the course of an in-vivo testing. This important step would allow to close the gap between the preclinical testing and the clinical entry of these modified and promising photosensitizers in particular in the difficult and challenging intraoperative application for malignant pleural mesothelioma.

To be honest I could not find any major query within the present manuscript and I want to wish good luck for the authors regarding their further research.

Reviewer 3 Report

            Bonsall and co-authors have presented the synthesis and application of new photosensitizers for photodynamic therapy (PDT) in mesothelioma cell lines. With the prominent rise in application of non-invasive PDT, the development for new photosensitizers that are more effective based on localization, limited toxicity, and efficiency of phototoxicity is necessary. The author present introduction of fatty acids to Foscan, a parent photosensitizer currently being employed for PDT, however still with some drawbacks. By altering the lengths of fatty acids conjugated to the porphyrin the authors looked to see the effect of hydrophobicity for PDT performance. Through their cell viability studies, the authors demonstrated that the novel pentanoic acid-conjugate (C5SHU), demonstrated the most efficient PDT effect of the three novel derivatives. While there is promise in the work the authors presented there are still issues that the authors should address before publication.

1: The major concern with the experimental setup is their use of light box. PDT is primarily applied utilizing red to near-IR light due to the deeper penetration through tissue than shorter wavelengths of light. This is mentioned in the introduction on page 2 line 64 but is never concreted stated. Although discussed later, page 18 line 415, Soret bands are rarely if ever used for PDT due to the limited light penetration, therefore there is little interest in photosensitizers that are only active at those wavelengths. When doing their light irradiation for cell viability assays the authors state on page 4 line 150 that, “plates were irradiated for 15 minutes on a 15W light box”. Was a light filter applied so only light longer than 600 nm was used for irradiation?

2: It is recommended to run the same experiments, photophysics and cell viability, for the parent Foscan. This would demonstrate the improvement that introducing these fatty acid-conjugation provides, whether it be localizability or phototoxicity enhancements.

3: It would be beneficial if the authors could perform more in depth photophysical studies to demonstrate the improvements by these photosensitizers. There is no data showing Beer’s law to determine molar absorptivity given in supporting information Singlet oxygen quantum yield would also be very beneficial to further support the promise of these photosensitizers.

The paper also has some writing errors, listed below:

Page 2, Line 59: chlorine, should be spelt chlorin

Page 13: Figure 8 caption does not fit figure provided, stating 24 and 48 hours when only showing 48 hours

Page 15: Figure 10 caption also does not fit figure provided, this is true of many figures

Page 17, Line 360: at is superscript, should only be “a” correct?

Page 19, Line 448: Format of reference is not lined up with other references

Supporting Information: Superscripts and subscripts are not formatted at all, should be done. No need to include MS data when high-res MS data is already provided. The NMRs are poorly presented, different formatting between spectra and no integration on NMR for C7 acid.

Round 2

Reviewer 1 Report

The authors have incorporated all the changes raised in the previous version of the manuscript and henceforth, I recommend this manuscript be published in this version. 

Reviewer 3 Report

I am pleased with the efforts put forth by the authors to edit this manuscript. The document reads much clearer and demonstrates the strong future their modifications could have on PDT photosensitizers in the future. My only edit would be a few typos on page 2 of the manuscript. I believe the authors meant to say Photofrin and Foscan on line 58, not Photo fin and Fosca.

Other than that the manuscript is ready to publish and congratulations to the authors on great work.